# Placental Metabolomics of Fetal Growth Restriction

**DOI:** 10.3390/metabo13020235

**Published:** 2023-02-04

**Authors:** Jacopo Troisi, Steven J. K. Symes, Martina Lombardi, Pierpaolo Cavallo, Angelo Colucci, Giovanni Scala, David C. Adair, Maurizio Guida, Sean M. Richards

**Affiliations:** 1Department of Medicine, Surgery and Dentistry, “Scuola Medica Salernitana”, University of Salerno, 84081 Baronissi, SA, Italy; 2Theoreo srl, Via degli Ulivi 3, 84090 Montecorvino Pugliano, SA, Italy; 3Department of Chemistry and Biology “A. Zambelli”, University of Salerno, 84084 Fisciano, SA, Italy; 4Department of Chemistry and Physics, University of Tennessee at Chattanooga, 615 McCallie Ave., Chattanooga, TN 37403, USA; 5Department of Obstetrics and Gynecology, Section on Maternal-Fetal Medicine, University of Tennessee College of Medicine, 979 East Third Street, Suite C-720, Chattanooga, TN 37403, USA; 6Department of Physics, University of Salerno, 84084 Fisciano, SA, Italy; 7Istituto Sistemi Complessi—Consiglio Nazionale delle Ricerche, 00185 Rome, RM, Italy; 8Department of Neurosciences and Reproductive Sciences, University of Naples Federico II, 80131 Naples, NA, Italy; 9Department of Biology, Geology and Environmental Sciences, University of Tennessee at Chattanooga, 615 McCallie Ave., Chattanooga, TN 37403, USA

**Keywords:** metabolomics, fetal growth restriction, small for gestational age, low birth weight

## Abstract

Fetal growth restriction is an obstetrical pathological condition that causes high neonatal mortality and morbidity. The mechanisms of its onset are not completely understood. Metabolites were extracted from 493 placentas from non-complicated pregnancies in Hamilton Country, TN (USA), and analyzed by gas chromatography–mass spectrometry (GC–MS). Newborns were classified according to raw fetal weight (low birth weight (LBW; <2500 g) and non-low birth weight (Non-LBW; >2500 g)), and according to the calculated birth weight centile as it relates to gestational age (small for gestational age (SGA), large for gestational age (LGA), and adequate for gestational age (AGA)). Mothers of LBW infants had a lower pre-pregnancy weight (66.2 ± 17.9 kg vs. 73.4 ± 21.3 kg, *p* < 0.0001), a lower body mass index (BMI) (25.27 ± 6.58 vs. 27.73 ± 7.83, *p* < 0.001), and a shorter gestation age (246.4 ± 24.0 days vs. 267.2 ± 19.4 days *p* < 0.001) compared with non-LBW. Marital status, tobacco use, and fetus sex affected birth weight centile classification according to gestational age. Multivariate statistical comparisons of the extracted metabolomes revealed that asparagine, aspartic acid, deoxyribose, erythritol, glycerophosphocholine, tyrosine, isoleucine, serine, and lactic acid were higher in both SGA and LBW placentas, while taurine, ethanolamine, β-hydroxybutyrate, and glycine were lower in both SGA and LBW. Several metabolic pathways are implicated in fetal growth restriction, including those related to the hypoxia response and amino-acid uptake and metabolism. Inflammatory pathways are also involved, suggesting that fetal growth restriction might share some mechanisms with preeclampsia.

## 1. Introduction

Low birth weight (LBW) is a condition characterized by a neonatal birth weight of less than 2500 g, occurring either in preterm or in full term. Small for gestational age (SGA) neonates are also LBW, but are more precisely defined according to gestational age. A low birth weight is correlated with many socioeconomic factors and parameters affecting intrauterine growth restriction (IUGR) and preterm birth.

LBW has a global incidence of 17% and is an important predictor of infant mortality [1]. LBW also increases the risk of chronic diseases in adults, such as heart disease and type 2 diabetes [2,3]. Birth weight is correlated with heart-metabolic risk at 5 years old. Lurbe et al. [4] reported that insulin and triglyceride serum levels at 5 years old depend on birth weight, current weight, and postnatal weight gain.

Maternal exposure to environmental pollutants occupies a central role in the genesis of LBW [5,6]. In Hamilton Country, TN, USA, there is a high rate of low birth weight infants when compared with other metropolitan areas in the state, as well as the national rate [7]. This phenomenon has been extensively studied in recent years, but little progress has been made in its understanding. Maternal habits and socio-economic factors appear to play a role, but it seems that they cannot fully explain the abnormally high incidence of LBW in Hamilton County.

An innovative approach for examining the complexities associated with LBW is the field of metabolomics, which is the study of the identities and abundances of the entire metabolome, which is the whole set of metabolites present in a biological sample at a given time [8,9]. Metabolites are the substrates or products of biochemical reactions within cells and tend to be small molecules weighing less than 1500 Daltons [9]. An individual’s metabolome is strongly influenced by interactions with the external environment and by the particular products of gene expression. Metabolomics differs from genomics, because it identifies and quantifies the products of specific protein biochemical reactions occurring within the cell, tissue, or organ as opposed to the genetic expression.

Although it is a powerful approach, metabolomics has only recently been applied to the investigation of LBW associations. In 2012, Kenny and Baker [10] earned a patent for their invention of a metabolomic approach to predict SGA infant development at a pre-symptomatic gestational stage, based on the analysis of maternal blood. Changes to the metabolome in the maternal serum during pregnancy were investigated by Heazell et al. [11] and Tea et al. [12]. Both reported changes in maternal energy metabolism associated with the LBW fetal condition. Umbilical cord blood was investigated by Ivorra et al. [13] and Alexandre-Gouabau et al. [14]. Both studies reported a change in amino acid metabolism that was associated with LBW infants. Furthermore, urine metabolomics of children [15] and adults [16] born with LBW found that various metabolites were associated with subclinical renal and cardiac injury. Horgan et al. [17] reported changes in the metabolic footprint of placentae cells of small for gestational age and normal birth weight pregnancies when explanted and cultured in different oxygen tensions.

Recent applications of metabolomics to investigate a variety of obstetrical complications include a study on polycystic ovarian syndrome [18] and the identification of metabolomic biomarkers for pre-delivery screening for developmental anomalies [19], chromosomal anomalies [20], central nervous system anomalies [21], heart defects [22], and endometrial cancer [23,24]. The present study consisted of a large population-based assessment that investigated the whole metabolome of postpartum placental tissue from women living in Hamilton Country, TN, USA, in order to better understand the placental metabolic pathway and its potential associations with SGA and LBW infants.

## 2. Materials and Methods

### 2.1. Tissue Collection

The University of Tennessee College of Medicine’s Institutional Review Board (IRB) approved the gathering and utilization of placentae for the present research (IRB#05-031, FWA#2301, 2005). The specimens were collected at Baroness Erlanger Campus of the Erlanger Health System, located in Chattanooga, TN, between June 2007 and July 2010. Sample collection was based on the timing of maternal arrival and placental delivery. Sample collection was only performed when investigators of this study could be present to ensure that standard collection procedures were followed. As this timing was somewhat random (subject to personnel availabilities), this reduced any possibility of a “timing-based” bias. Further information on mothers and infants was obtained from medical records. The placentae were obtained in a standardized manner from single birth mothers over 18 years old, who were free of HIV and hepatitis. To avoid analytical bias, all of the samples were collected, stored, processed, and analyzed in the exact same way. The whole placentae were stored at −80 °C until they were processed. Prior to the analysis, samples were thawed overnight at room temperature and then the chorionic plate and decidua basalis were removed with ceramic scissors, leaving the villous core tissue to be homogenized. A portion of the homogenized villous core was taken for the metabolomic analysis. Samples were oven-dried at 60 °C to constant weight and then powderized with a mortar and pestle.

### 2.2. Birth Weight Centile Calculation

In this study, two key factors were analyzed: raw birth weight and the calculated birth weight percentile. The obstetrical community considers infants with a weight less than 2500 g as being low birth weight (LBW). The birth weight percentile was calculated for each sample using the gestation related optimal weight (GROW) software [25]. This calculation considers various parameters that influence birth weight, such as gestational age, maternal height/weight, ethnicity, parity, and infant sex. The GROW software is dependent on the geographic location; thus, it employs nationwide data to determine specific coefficients for each growth parameter. This study used the United States version of GROW. The data for GROW and the birth weight percentile calculation were obtained from the anonymous medical charts linked to each placenta. The gestational age was determined from the last menstrual period and was based on the best obstetric estimate from either the first or second trimester ultrasound examinations. Neither dietary nor socioeconomic data were collected for this study. Based on the birth weight percentile, newborns were classified into three categories: small for gestational age (SGA) (<10%), appropriate for gestational age (AGA) (10–89%), or large for gestational age (LGA) (>90%).

### 2.3. Metabolite Extraction, Purification and Derivatization

The MetaboPrep GC kit from Theoreo was used for untargeted metabolome extraction, purification, and derivatization. Sample analysis was performed in the period from December 2016 to August 2017. Following the manufacturer’s instructions, 25 ± 1 mg of tissue was added to an Eppendorf microcentrifuge tube along with the extraction mixture, including the internal standard (2-isopropyl malic acid). The sample and extraction mixture were mixed via vortexing at 1250 rpm for 30 min and then subjected to ultrasonic treatment (30 min at 30 °C). The extracted metabolome was centrifuged to separate any solids before the purification process. The purified sample was then freeze-dried overnight.

Trimethylsilyl derivatization was carried out by adding the first derivatization mixture and vortexing, followed by the second derivatization mixture. The derivatized metabolomes were transferred to 100 µL inserts for the auto sampler injection. Before injection into the GC–MS, the samples were centrifuged.

### 2.4. GC–MS Analysis

A sample of the derivatized solution (2 microliters) was injected into a GC–MS system (GC-2010 Plus gas chromatograph with a 2010 Plus single quadrupole mass spectrometer by Shimadzu Corp. in Kyoto, Japan) using split mode. Chromatography was performed with a 30 m × 0.25 mm CP-Sil 8 CB fused silica capillary GC column with a 1.00 µm film thickness from Agilent, using helium as the carrier gas.

The oven temperature program consisted of a 0.5-min hold at 100 °C and was then increased to 320 °C at 4 °C/min, resulting in a GC run time of 60 min. Carrier gas flow linear velocity was maintained at 39 cm/s and the split flow was set to 5:1. The mass spectrometer was programmed with a 4.5 min solvent delay and then operated in full scan mode (35–600 m/z) using electron ionization (70 eV) and a scan velocity of 3333 amu/s.

The samples were analyzed in groups of 25, with four controls per batch: a blank injection of hexane, a standard mix of 50 molecules, a pool of 2 microliters from 50 randomly selected treated samples, and a repeated injection of one randomly selected sample. Each batch was deemed valid if the blank showed no peaks, the standard peak area was within 10% of what was expected, the ratio of the 100 major peaks in the repeated sample was within 15% of the original, and the pooled sample was within 5% of the model built from previous samples.

The chromatograms were deconvoluted using the most intense fragment for metabolites meeting the integration criteria (area > 10000; slope > 100/min; width > 1 s). 2-isopropyl malic acid was used as the internal standard (SIM = 147) and the extracted chromatograms were used to calculate the metabolite area. Relevant metabolites (see below) were annotated according to level 1 of the Metabolomics Standard Initiative [26]. Briefly, after mass spectra comparison with several public libraries (NIST2014 and HMDB), and also using the Kovats retention index [27] to reduce the candidate list, an analytical standard was derivatized and analyzed in the same condition to confirm the peak annotation.

### 2.5. Statistical Analysis

#### 2.5.1. Demographics Data Comparison

The clinical and demographic data were not normally distributed according to the Kolmogorov–Smirnov test, so that only non-parametric statistical tests were employed for a comparison of these characteristics. The comparison between groups was made with the Rank Sum Test, according to Mann–Whitney or by means of analysis of variance on rank preformed single way (according to Kruskal–Wallis), also using the post-hoc test of Dunn (*p* < 0.05). Comparisons of variables reported as percentages were done by means of the Yates correction of the Χ^2^ test.

#### 2.5.2. Metabolomic Data Analysis

Partial least square discriminant analysis (PLS-DA) was performed using the R statistical platform (version 4.1.2; http://www.R-project.org accessed on 10 January 2022; R-Study version 1.4.1717 was used as IDE). Chromatographic data pre-treatment included chromatogram alignment, calculation of peak areas normalized to that of the internal standard, followed by mean centering and unit variance scaling (i.e., auto-scaling), and were conducted using the MetaboPredict software (version 1.2.2, Theoreo srl, Montecorvino Pugliano (SA), Italy). Class separation was investigated by PLS-DA, which is a supervised method that uses multivariate regression techniques to extract, via linear combination of the original variables (X), the information that can predict the class membership (Y). Variable importance in projection (VIP) scores were calculated for each component. Features with VIP score ≥ 2 were identified for further biochemical pathway investigation. Lower scores were not considered to be significant contributors to class separation. Classification and cross-validation were performed as well as a permutation test to assess the significance of the class discrimination and the lack of model overfitting.

The PLS regression was carried out using the plsr function provided by the pls package in R [28]. Classification and cross-validation were performed with the wrapper function provided in the caret package in R [29]. To gauge the significance of class differentiation, a permutation test was conducted. During each permutation, a PLS-DA model was established by cross-validating the optimal number of components between the data (X) and the permuted class labels (Y). Two forms of test statistics were calculated to measure the class differentiation. The first was based on training prediction accuracy, while the second was a separation distance based on the B/W ratio (ratio between group sum of squares and within group sum of squares). If the observed test statistics were part of the distribution produced from the permuted class assignments, class differentiation was not considered statistically significant [30].

Variable importance in projection (VIP) scores were calculated for each component. VIP is a weighted sum of the PLS loadings’ squares, considering the amount of Y-variation explained in each dimension. The weights are determined by the reduction in the sums of the squares across the number of PLS components. The average of the feature coefficients was used to indicate the overall coefficient-based importance.

The involvement of metabolic pathways was analyzed using the MetPa tool [31]. It is based on an over-representation analysis test. If a particular group of compounds is more represented than expected by chance within the compound list, this pathway was selected. The over-representation was based on Fisher’s Exact test. The over-representation analysis was performed using only statistically relevant metabolites (i.e., those with VIP-scores > 2 and those selected based on the volcano plot) as inputs for the MetPa analysis.

## 3. Results

Results were obtained from 493 placental tissues, 124 from pregnancies with LBW newborns and 369 from pregnancies resulting in birth weight >2500 g. Newborns were determined to be SGA in 118 cases, AGA in 326, and LGA in 49 by normographic data. The demographic and clinical characteristics of all pregnancies are reported in Table 1. 

Women who had an LBW newborn also had the following characteristics: lower pre-pregnancy weight (66.2 ± 17.9 kg vs. 73.4 ± 21.3 kg, *p* < 0.0001), a lower body mass index (BMI) (25.27 ± 6.58 vs. 27.73 ± 7.83, *p* < 0.001), and a shorter gestation time (246.4 ± 24.0 days vs. 267.2 ± 19.4 days *p* < 0.001) when compared with non-LBW subjects. Marital status, tobacco use, and fetus sex affected the birth weight centile classification according to gestational age. For example, single mothers had an LGA infant more frequently than an AGA infant (69.4% vs. 53.1%, *p* < 0.05) and the LGA infant was more frequently male (65.3% vs. 50.0%, *p* < 0.05). A higher tobacco use frequency was associated with SGA infants compared with AGA infants (31.4% vs. 17.2%, *p* < 0.05).

Gas chromatography–mass spectrometry consistently detected 272 endogenous metabolites in each specimen. These compounds are involved in many biochemical processes, such as energy metabolism, lipid metabolism, and amino acid metabolism. For chromatographic peak identification, the linear retention index difference max tolerance was set to 10, while the minimum matching for the NIST library search of the corresponding mass spectrum was set to 85%. The results were summarized in a comma separated matrix file and loaded in the appropriate software for statistical manipulation.

After data alignment using the Parametric Time Wrapping algorithm [32] and peak picking, integration, and deconvolution, the chromatographic data were tabulated with one sample per row and one variable (metabolite) per column. The normalization procedures consisted of data transformation and scaling. Data transformation was generalized log transformation while data scaling was auto scaling (mean-centered and divided by standard deviation of each variable) [33]. Two PLS-DA models were considered (Figure 1); a two-class model (LBW vs. Non-LBW; panel A) and a three-class model (SGA, AGA, and LGA; panel B).

A well-defined differentiation of the LBW and non-LBW was achieved (R^2^_Y_cum = 0.82, Q^2^_Y_cum = 0.59) (Figure 1-A1). Discrimination among SGA, AGA, and LGA was more robust with less overlap (R^2^_Y_cum = 0.85, Q^2^_Y_cum = 0.71) (Figure 1-B1). Variable importance in projection (VIP) scores were calculated for each component in the PLS-DA regressions. Panels A2 and B2 of Figure 1 show the metabolites selected as being those most responsible for class separation (with a VIP-score > 2) for the two-class and three-class models, respectively.

Five metabolites were selected as being relevant by the volcano plot: hydroquinone and hydroxylamine were lower in LBW, whereas deoxyguanosine, glutathione, and linoleic acid were higher in LBW (Figure 2).

Metabolites with *p* < 0.05 assessed by the ANOVA of the three classes were also evaluated in a heatmap. The consistent selection of relevant metabolites using the different models was synthesized with an UpSet visualization [34] showing that glycerophosphocholine, tyrosine, and serine were selected using three tools (heatmap, 2- and 3- classes PLS-DA); deoxyguanosine, hydroquinone, and linoleic acid were selected by heatmap and volcano; and asparagine, glycine, taurine, isoleucine, aspartic acid, and deoxyribose were selected by both PLS-DA models. All of the other metabolites were selected by only one model (Figure 2). VIP metabolites concentration distribution in the studied classes are also reported in Figure 3.

Figure 4 shows the metabolic maps which better explain the interplay of the selected metabolites. Several metabolic pathways are involved, including the glycerophsphate shuttle; ammonia recycling; and glutathione, glycerolipid, tyrosine, glutamate, and phospholipid metabolism.

## 4. Discussion

In the present study, we report the placental metabolomic fingerprint of LBW and SGA newborns, a subject of growing interest due to the recent importance given to the concept of “fetal origin of adult disease”. This idea suggests that early life conditions can “program” the fetus for a spectrum of adverse health outcomes in adulthood, such as coronary artery disease, hypertension, obesity, and insulin resistance [35,36]. Indeed, for the first time in human history, a generation of LBW and VLBW (very low birth weight < 1000 g) infants have been identified and monitored until adulthood [37,38], which supported the theory of “fetal origin of adult disease”. However, we are still far from understanding the long-term effects of low birth weight.

Although the placenta is the organ mainly involved in fetal nutrition and weight determination, few studies have investigated the placental metabolomic variations associated with newborn weight [17,39]. The results obtained herein indicate that several metabolites and metabolic pathways are associated with a low-birth-weight placental metabolic fingerprint. The network of tissue molecules allowing for class separation was characterized by lower levels of taurine, ethanolamine, β-hydroxybutyrate, and glycine and higher levels of many amino acids (asparagine, aspartic acid, isoleucine, serine, and tyrosine), deoxyribose, erythritol, glycerophosphocholine, and lactic acid in the SGA/LBW placentas we reported.

Taurine is an essential nutrient in fetal metabolism, as the fetus and placenta lack the enzyme for taurine synthesis [40]. Reduced activity of a placental taurine transporter has been found in preeclampsia, a condition frequently associated with fetal weight restriction [41]. Reduced taurine in the placenta may impair syncytiotrophoblast cell renewal and lead to decreased nutrient transfer to the fetus [41]. Austdal et al. [42] reported that taurine levels were similar in the placentae from preeclampsia with or without fetal growth restriction, suggesting that taurine depletion is not specific for fetal growth restriction, but is probably associated with an impairment in placental mechanisms. Ethanolamine is a component of phospholipid biosynthesis. Murine experiments indicate that ethanolamine kinase deficiency results in low birth weight offspring and increased placental thrombosis and apoptosis, indicating an important role of ethanolamine in placental and fetal development [43]. Deoxyribose is a reducing sugar with angiogenic properties [44]. It has been shown to alter apoptosis and glutathione expression in vitro [45]. Its increase has been reported in placental tissue cultivated under hypoxic conditions [11]. It can be released from cells, either by active transport from living tissue or from dead cells following lysis. Accordingly, an increase in erythritol, a sugar alcohol derived either from external sources or from the reduction of erythrose, was also reported in placental tissue and in a culture medium under hypoxic condition [11]. Increased levels of erythritol were reported during heart failure [46], indicating a possible role in the hypoxic response. In addition, Arkwright et al. [47] described a differential expression of small sugars and their derivatives in the syncytiotrophoblast glycocalyx in preeclampsia pregnancies. Such increases in sugar alcohols may indicate altered glucose metabolism, which Kay et al. [48] previously showed to be modified in trophoblast in response to changes in O_2_ tension.

Two possible reasons for increased glycerophosphocholine in LBW/SGA placentas can be suggested. First, the increase may be due to increased cell death [49]. Second, the increase may stem from placental cell membrane catabolism for the regeneration of choline methyl groups due to folate deficiency [50].

Amino acids are among the main nutrient sources for fetal growth, accounting for 20–40% of fetal energy requirements [51]. Amino acids are actively transported by the placenta from the maternal to the fetal circulation [52]. Indeed, the placental and fetal uptake of amino acids is in excess with respect to fetal protein synthesis needs [52]. Our report of a higher concentration of asparagine, aspartic acid, isoleucine, serine, and tyrosine in SGA/LBW placentas can be the result of dysregulation in the placental amino acid transport system. Fetal amino acid supply is subjected to a maternally regulated transport system. Maternal signals that provide information to the placenta include metabolic hormones, nutrients levels, and oxygen. In conditions of compromised maternal supply line, the ability to deliver nutrients and oxygen to the placenta and placental functions, including transplacental nutrient transport and placental growth, may be inhibited, directly contributing to decreased fetal growth [53]. This transport system seems to be regulated by the mTOR (mammalian target of rapamicine) sensing system [54], a system of considerable interest for its implication in aging and commonly associated with several diseases.

The glycine transport system (called System A) is an Na-dependent system and is concentrated in the syncytiotrophoblast, energized by the Na^+^ gradient. Subsequently, glycine diffuses into the fetal circulation and back to the mother. The strong polarization of System A provides the basis for the net transport of glycine to the fetus. Paolini et al. [55] reports that System A transport is impaired differently compared with the leucine and tyrosine transport system when associated with fetal weight restriction. It may be that the leucine and tyrosine transport system is not as susceptible to low oxygen tension and high concentrations of lactic acid such as those observed in the LBW/SGA placentae herein. If so, this would help to explain the different prevalence of glycine compared with the other VIP amino acids that were increased in LBW.

Our observation of increased lactic acid concentration in LBW/SGA placentae strongly indicates hypoxic conditions of the placental cells. In this condition, ketone utilization was increased. In murine models, hypoxic tolerance was increased in those mice treated with 3-hydroxybutyrate [56]. This finding provides some evidence to explain the lower concentration of β-hydroxybutyrate found in the LBW/SGA placentas.

Women with SGA babies showed higher self-reported tobacco use. One of the main ways in which smoking affects fetal growth is through the reduction of oxygen and nutrients that reach the developing baby. Moreover, nicotine, a component of tobacco smoke, is a vasoconstrictor, which means it narrows the blood vessels and decreases blood flow. This reduction in blood flow can result in a decrease in oxygen and nutrients reaching the placenta, which can lead to SGA babies. Additionally, smoking increases the risk of placental abruption, where the placenta separates from the uterus prematurely, cutting off the baby’s oxygen supply and increasing the risk of stillbirth. Another way in which smoking can affect fetal growth is through the release of harmful chemicals from tobacco smoke. These chemicals can cross the placenta and reach the developing baby, causing damage to the developing cells and organs. For example, carbon monoxide, a component of tobacco smoke, can reduce the amount of oxygen in the baby’s blood, leading to fetal growth restriction. The risks associated with smoking during pregnancy are not limited to women who smoke cigarettes. Secondhand smoke exposure can also have negative effects on fetal growth and development.

Nevertheless, the use of tobacco is not associated with the reported differences in terms of metabolomics profiling within our cohort. Indeed, the abundances of VIP-score and Volcano plot selected metabolites did not show significant differences in smoker and non-smoker enrolled subjects (data not shown). The same evaluations were performed for marital status, education, and infant sex; the abundances of statistically significant metabolites were also not correlated with these parameters.

The placental metabolic profile of LBW/SGA newborns highlights the role of amino acid metabolism and degradation, and the role of folate, prostaglandin, and leukotriene metabolism. This illustrates the great importance of hypoxic and nutritional placental deficiency, but also suggests a possible role of inflammation, which may be responsible for a subclinical preeclampsia condition.

The present study used a novel placental analysis technique and contributes to previous metabolomic studies carried out on cultured placental explants. Indeed, while oven drying is not standard in human metabolomics investigations, it is widely used in plant metabolomics [57] with no adverse effects. Complete water removal from tissue prior to metabolite extraction is necessary because water provides the medium for enzyme-mediated reactions, which could adversely affect the resulting metabolomes if metabolite decomposition occurs. Of course, varying levels of water could then affect the overall pattern and concentration of metabolites [57], underscoring the importance of complete water removal. Moreover, Troisi et al. [58] reported that oven-drying is an easy and affordable method for placenta sample preparation. Oven-dried placenta metabolomes were able to distinguish complicated vs. non complicated pregnancies in the same manner that fresh and lyophilized placenta samples could [58]. Notably, as Roessner et al. [59] observed, regardless of which sample preparation techniques are employed, biological variability is often greater than analytical variability. Most importantly for the relative comparisons described herein, because all of the samples were treated identically, the statistical testing and conclusions drawn were internally consistent.

Our method provides a unique tool for understanding placental disease mechanisms in utero, thereby phenotyping diseases based on the whole placental involvement and not on results from cultured cells. Because these kinds of studies require a large cohort to provide accurate and reproducible findings, our strength is that our cohort is one of the biggest ever used in a metabolomics study.

## Figures and Tables

**Figure 1 metabolites-13-00235-f001:**
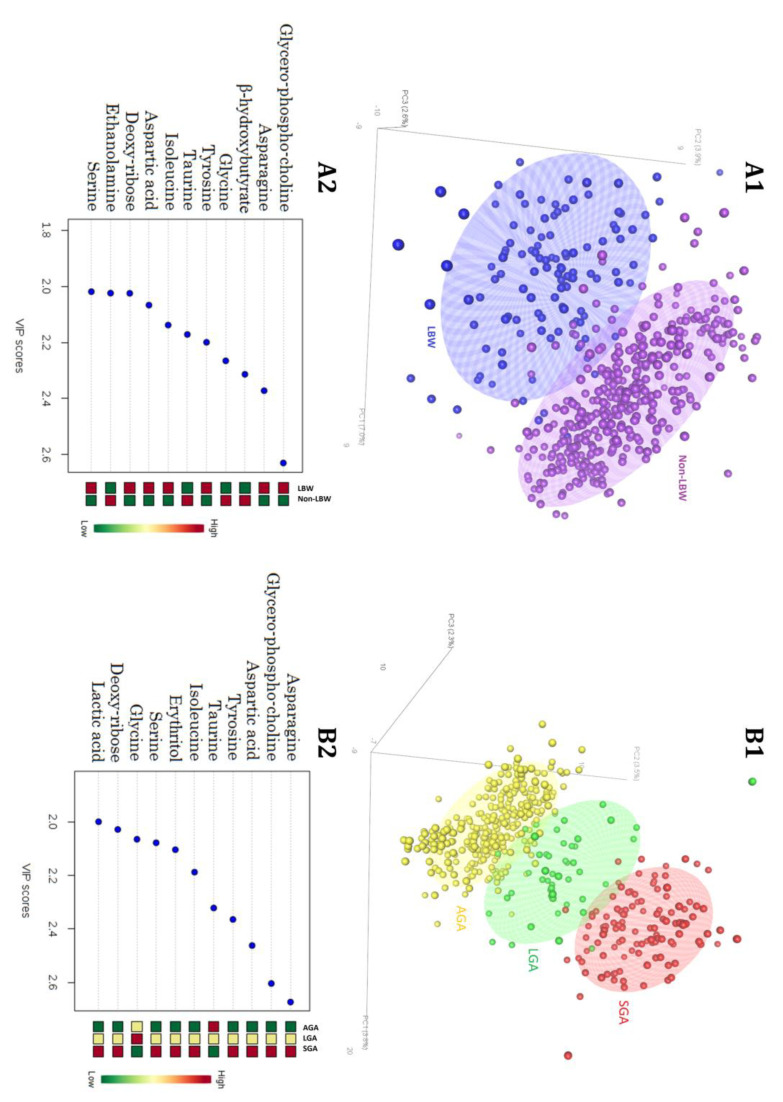
Classification models: (**A1**). Two class (LBW vs. non-LBW) PLS-DA score plot. (**A2**). VIP metabolites heat-map selected by the dichotomic classification. (**B1**) Three class (SGA, AGA, and LGA) PLS-DA score plot. (**B2**) VIP metabolites heat-map selected by the three-class classification. Abbreviations: LBW: low birth weight; PLS-DA: partial least square discriminant analysis; VIP: variable important in projection; SGA: small for gestational age; AGA: adequate for gestational age; LGA: large for gestational age.

**Figure 2 metabolites-13-00235-f002:**
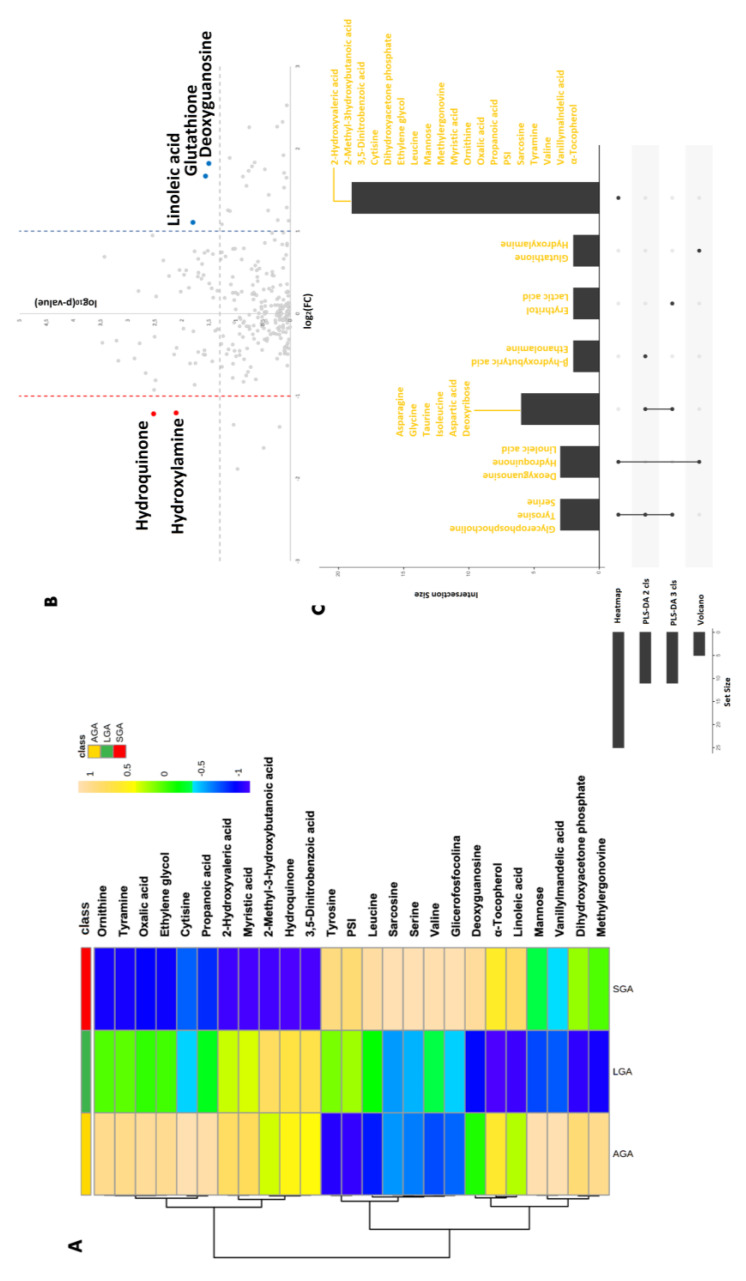
Relevant metabolites selected in all class comparisons. (**A**) Heatmap reporting the ANOVA selected metabolites. Cluster analysis allowed for the recognition of three groups of metabolites based on their concentration mean levels in the three classes (SGA: small for gestational age; AGA: adequate for gestational age; LGA: large for gestational age). (**B**) Volcano plot of the analyzed metabolites comparing LBW to non-LBW placentae, highlighting the ones showing a *p*-value < 0.05 and a large fold change (>2.0 or <0.5); in red are the metabolites with a lower concentration in LBW, while in blue are the metabolites with a higher concentration in LBW. (**C**). UpSet diagram reporting the relevant metabolite selection among the various strategies. Set size indicates the number of metabolites involved in each metabolite selection strategy, while intersection size indicates the number of metabolites selected from each strategy combination.

**Figure 3 metabolites-13-00235-f003:**
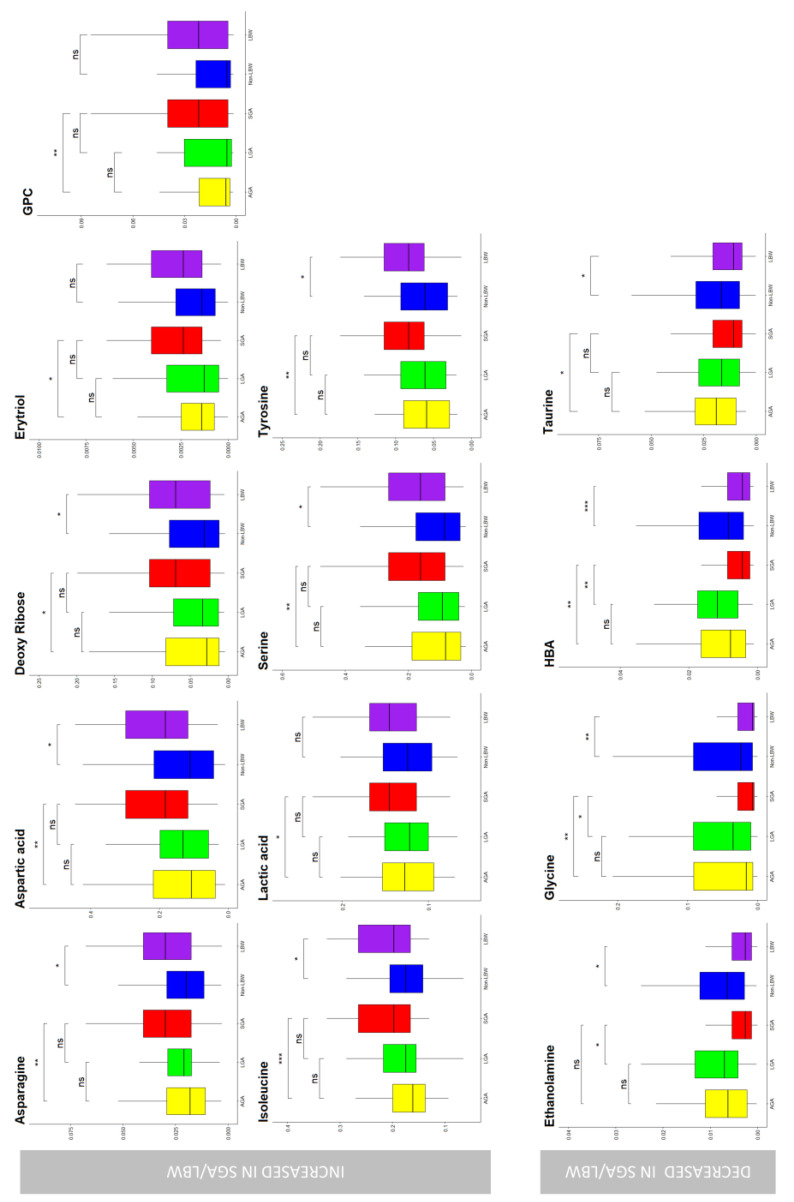
Box and whisker plot of the VIP metabolites. Red boxes represent the placentae of small for gestational age (SGA) newborns (*n* = 118), yellow represents the adequate for gestational age (AGA) placentae (*n* = 326), green represents the large for gestational age (LGA) placentae (*n* = 49), blue represents the placentae of low birth weight baby (LBW) (*n* = 124), while purple represents the non-low birth weight baby (non-LBW) (*n* = 369). The y axes represent the normalized metabolite chromatographic area. GPC = Glycerophosphocoline; HBA = Hydroxybutyric acid; ns = not significance. Ns indicated not significant difference in concentration, * *p*-value < 0.05, ** *p*-value < 0.01, *** *p*-value < 0.001.

**Figure 4 metabolites-13-00235-f004:**
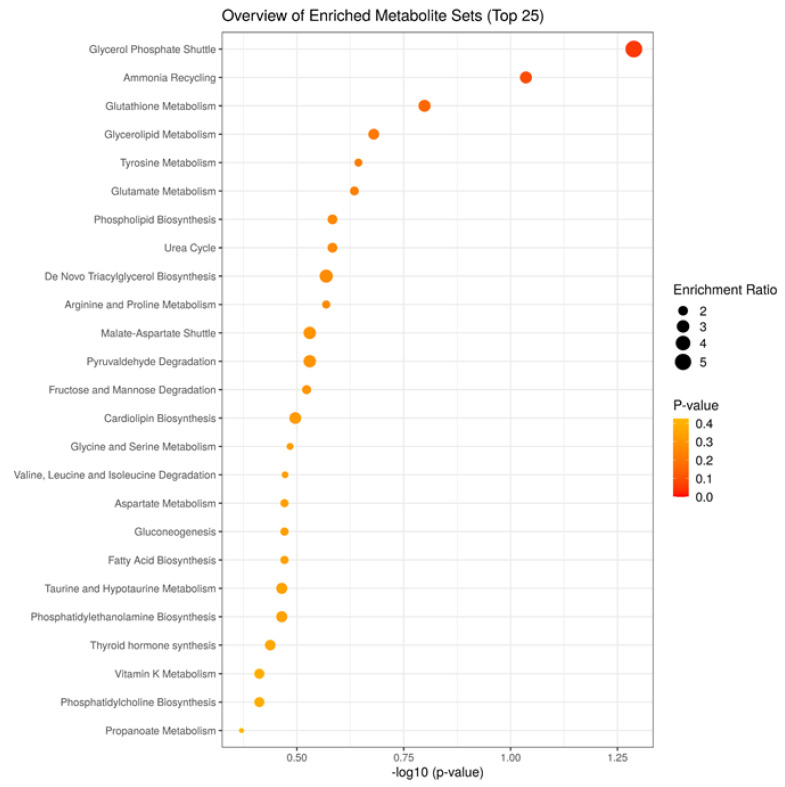
Metabolic pathways showing the interplay of selected metabolites.

**Table 1 metabolites-13-00235-t001:** Population characteristics of the cohort profiled in the present untargeted metabolome investigation. * Indicates a significant difference from Non-LBW. ¶ indicates a significant difference from AGA. Abbreviations. BMI: Body mass index, HS/GED: High school/General Educational Development, LBW: low birth weight; SGA: small for gestational age; AGA: adequate for gestational age; LGA: large for gestational age.

	All Patients	LBW	Non-LBW	SGA	AGA	LGA
Sample size	493	124 (25.2%)	369 (74.8%)	118 (23.9%)	326 (66.1%)	49 (9.9%)
Age (years)	26.0 ± 5.5	25.4 ± 5.6	26.2 ± 5.5	25.6 ± 5.8	25.9 ± 5.4	27.8 ± 5.6
Height (cm)	162.6 ± 10.4	162.3 ± 14.5	162.7 ± 8.6	163.6 ± 7.8	162.0 ± 8.9	164.8 ± 20.4
Weight before pregnancy (kg)	71.6 ± 20.7	66.2 ± 17.9 *	73.4 ± 21.3	73.3 ± 24.3	70.5 ± 19.2	74.6 ± 20.5
BMI	27.7 ± 7.6	25.3 ± 6.6 *	27.7 ± 7.8	27.3 ± 8.5	26.9 ± 7.2	28.0 ± 7.9
Underweight (<19)	7.5%	12.9%	5.7%	9.3%	7.1%	6.1%
Normal weight (19–25)	38.1%	42.7%	36.6%	38.1%	39.0%	32.7%
Overweight (25–30)	27.0%	21.0%	29.0%	32.2%	27.6%	30.6%
Obese (>30)	27.4%	23.4%	28.7%	20.3%	26.4%	30.6%
Marital Status						
Single	52.5%	41.1%	56.4%	44.1%	53.1%	69.4%¶
Married	47.5%	58.9%	43.6%	55.9%	46.9%	30.6%¶
Race						
White	63.5%	58.9%	58.2%	63.6%	63.2%	65.3%
Black	19.9%	25.8%	17.9%	24.6%	18.7%	16.3%
Hispanic	16.2%	15.3%	23.9%	11.9%	17.8%	16.3%
Other	0.4%	0.0%	0.0%	0.0%	0.3%	2.0%
Education						
<HS	32.5%	32.7%	32.4%	27.1%	36.0%	21.2%
HS/GED	30.9%	33.6%	29.7%	40.6%	27.3%	30.3%
College	36.6%	33.6%	37.9%	32.3%	36.8%	48.5%
Tobacco use						
No	81.1%	75.0%	82.4%	68.6%¶	82.8%	93.9%
Yes	18.9%	25.0%	17.6%	31.4%¶	17.2%	6.1%
Parity	1.3 ± 1.8	1.1 ± 1.4	1.4 ± 1.9	1.5 ± 3.0	1.2 ± 1.2	1.5 ± 1.3
Gestational age (day)	261.9 ± 22.5	246.4 ± 24.0 *	267.2 ± 19.4	260.0 ± 26.0	263.8 ± 17.8	254.1 ± 36.3¶
Infant sex						
Male	52.1%	54.8%	47.7%	52.5%	50.0%	65.3%¶
Female	47.9%	45.2%	52.3%	47.5%	50.0%	34.7%¶

## Data Availability

The data that support the findings of this study are available from the corresponding author upon reasonable request. The data are not publicly available due to privacy.

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
