# Peer review of "Placental Metabolomics of Fetal Growth Restriction"

_metabolites, 2023, doi:10.3390/metabo13020235_

Round 1

Reviewer 1 Report

The paper provides crucial and interesting information about placental metabolomics. The study is conducted on a large population.

Nevertheless, I have some small suggestions about the paper:

1.       The placentas were collected from 2007 to 2010, so the biological material is quite old, even if were stored at -80℃. Or measurements were conducted earlier and now only published?

2.       When the authors received the consent of Bioethical Committee? The year is missed.

3.       In my opinion Material section should be introduce with characteristic of women and table 1 should be as description of Material instead results.

4.       The results should be also evaluate according to exposure to tobacco smoke, because smoking is one of the major factor affected fetal and its development

5.       Which exactly part of placenta was taken into the study? In all cases it was the same part?

6.       Small errors were found: example “verse 45: “socioeconomic” or “socio-economic” please use this word in one way; verse 352 “in vitro” should be written using italic style

7.       In my opinion lines 105-112 should be moved into discussion section instead Materials & methods section

8.       In my opinion discussion section should be more focused on results and their consequences, especially when authors would like to claim that their results indicate the crucial information about “fetal origin of adult disease”. For me it is too far-reaching proposal.

9.       In verse 330 please give more information about the type of those diseases

Author Response

Dear reviewer,

thank you for your work on our manuscript and for the interesting and useful suggestions for improvement.

Please find below a list of responses to all the reported concerns.

  1. The placentas were collected from 2007 to 2010, so the biological material is quite old, even if were stored at -80℃. Or measurements were conducted earlier and now only published?

Reply: Analysis were during 2016/2017.We added this detail in material and method section. There is international agreement that -80 °C storage stops all enzymatic activity and therefore quenches the metabolic processing.

  1. When the authors received the consent of Bioethical Committee? The year is missed.

Reply: 2005.  The date has been added behind the IRB number.

  1. In my opinion Material section should be introduce with characteristic of women and table 1 should be as description of Material instead results.

Reply: We appreciated the suggestion, but the enrolled subject’s characteristic is a result of the analysis of the enrolled patients data. So, we think it is more correctly positioned in results section while in material and methods where we described the tissue collection.

  1. The results should be also evaluate according to exposure to tobacco smoke, because smoking is one of the major factor affected fetal and its development

Reply: We appreciated the suggestion. We evaluated the relevant metabolites concentration distribution in the women which reported tobacco use and those that did not. No statistical differences emerged. We reported this evidence in the discussion section.

  1. Which exactly part of placenta was taken into the study? In all cases it was the same part?

Reply: As stated on line 104:  Prior to analysis, samples were thawed overnight at room temperature and then the chorionic plate and decidua basalis were removed with ceramic scissors, leaving the entire villous core tissue to be homogenized. To further clarify, the following statement was added following the above statement: A portion of the homogenized villous core was taken for metabolomic analysis.

  1. Small errors were found: example “verse 45: “socioeconomic” or “socio-economic” please use this word in one way; verse 352 “in vitro” should be written using italic style

Reply: Thanks. Fixed

  1. In my opinion lines 105-112 should be moved into discussion section instead Materials & methods section

Reply: Moved, please see line 407-422

  1. In my opinion discussion section should be more focused on results and their consequences, especially when authors would like to claim that their results indicate the crucial information about “fetal origin of adult disease”. For me it is too far-reaching proposal.

Reply: We are happy to address specific comments, however, this comment is too vague.  We are not sure if the reviewer does not agree with the concept of fetal origin of adult disease, or if there are other potential links related to metabolomics that the reviewer does not agree with.  The fetal origin of disease is a well-established theory (as cited) for which we are simply stating that metabolomics could be used to help to reduce the uncertainty.  We do not feel that such a connection is far-reaching.   The data produced by metabolomics could indeed help us to better vet the theory of fetal origin of adult disease.

  1. In verse 330 please give more information about the type of those diseases

Reply: This comment confuses us.  In our interpretation, this comment contrasts with the previous comment.  Regardless, the types of diseases associated with fetal origin of adult disease are detailed in the citations provided (36-39).  These disease are: coronary artery disease, hypertension, obesity, and insulin resistance. We have added this information to the paragraph.

Reviewer 2 Report

This manuscript, “Placental metabolomics of fetal growth restriction”, aimed to investigate the pathophysiological mechanism of fetal growth restriction by profiling the placental metabolome. Overall, the data analysis was appropriate, and the findings provided helpful insights.

The following comments or suggestions, if can be addressed, would further strengthen this manuscript.

  1. The samples were collected between 2007 and 2010, more than a decade ago. Does long-term storage affect certain metabolites that are more easily degraded?
  2. The author mentioned “Sample selection was based on maternal arrival time and time of placental delivery”. Could the author explain a little more how the samples were selected? Will the selection lead to potential selection bias?
  3. Suction title “2.3. Metabolite extraction, derivatization and derivatization” should be “2.3. Metabolite extraction, purification and derivatization”.
  4. Table 1 should be revised so that all the columns are aligned.
  5. Marital status, education, tobacco use, and SES etc. can be potential confounders. Authors should properly adjust these potential confounders and/or perform sensitivity analyses.
  6. How were metabolites annotated? The authors reported several metabolites of interest (glycerophosphorylcholine, tyrosine, serine, etc.) and it may be helpful if the authors could include GCMS mass spectra of these chemicals in their experiments along with reference spectra.
  7. What are the metabolites used as input for pathway over-representation analysis? Did authors adjust for background metabolites?
  8. The current resolution for Figure 2 is too low.

Author Response

Dear reviewer,

thank you for your work on our manuscript and for the interesting and useful suggestions for improvement.

Please find below a list of responses to all the reported concerns.

  1. The samples were collected between 2007 and 2010, more than a decade ago. Does long-term storage affect certain metabolites that are more easily degraded?

Reply: Samples were analyzed in 2016/2017. We reported this in par. 2.3. There is international agreement that -80 °C storage stops all enzymatic activity and therefore quenches the metabolic processing.

  1. The author mentioned “Sample selection was based on maternal arrival time and time of placental delivery”. Could the author explain a little more how the samples were selected? Will the selection lead to potential selection bias?

Reply: The sample collection was based on the ability to collect the placenta.  Because the placenta is routinely and quickly disposed of, we were only able to collect placentae when investigators were able to be present and properly store the placentae at -80 °C. We understand that this could be a source of bias; although such bias potential was reduced by having all samples collected, stored, processed, and analyzed in the exact same way.  Given the objective of the study and the large sample size, we believe that the differences we detected are representative of the population. See text for clarification.

  1. Suction title “2.3. Metabolite extraction, derivatization and derivatization” should be “2.3. Metabolite extraction, purification and derivatization”.

Reply: Thanks. Fixed

  1. Table 1 should be revised so that all the columns are aligned.

Reply: We designed the table in such a way to better allow the discrimination of principal subjects’ characteristics and subtype of choice for any of them. Anyway, editorial office will correct it during the proof producing.

  1. Marital status, education, tobacco use, and SES etc. can be potential confounders. Authors should properly adjust these potential confounders and/or perform sensitivity analyses.

Reply: We reported the evaluation of the selected metabolites among these subjects’ features. Please see line 381-401

  1. How were metabolites annotated? The authors reported several metabolites of interest (glycerophosphorylcholine, tyrosine, serine, etc.) and it may be helpful if the authors could include GCMS mass spectra of these chemicals in their experiments along with reference spectra.

Reply: Metabolites were annotated according to the level 1 MSI. Briefly after a mass spectra comparison with several public library (NIST2014 and HMDB), also using the Kovats index  to reduce the candidate list, analytical standard were derivatized and analyzed in the same condition to confirm the peak annotation. We reported this information in the MS. Please see line 164-168

  1. What are the metabolites used as input for pathway over-representation analysis? Did authors adjust for background metabolites?

Reply: All the relevant metabolites (selected by means of VIP-score and Volcano plot) were used as input list for the over-representation analysis. We added this statement in the MS, please see line 214-216

  1. The current resolution for Figure 2 is too low.

Reply: We uploaded high resolution image and increased their size in the MS.

Reviewer 3 Report

Dear Authors, your paper is interesting, important and almost clear.

I miss a brief statement about the transport of the tissues over the ocean.

See further details in the list below.

Details, typos, suggestions for improvements, and discussions

1. Lines 33, 35, you say ‘higher’ and ‘lower’. Missing is here: higher to what? and ‘lower than what?
The reason for this remark is that the reader does not necessarily have your prior knowledge on the matter, so she/he does not know your internal mental comparison.

2.       Line 132, has a blank between the less than sign and the 10% percentage. This is inconsistent with the other use of this symbol.

3.       Lines 205 – 229 do miss illustrations from the research application. To which variables did you apply the principal component analysis?  Currently, you only have a general explanation here which does not help to understand to what you apply it. You do not mention which classes were due to overfitting, for instance. For instance: you could refer here, or put here part of your PLS-DA explanation from lines 298 and further.

4.       Line 213, is unclear. ‘… using plsr function provided by R pls package [29].’  Should read ‘using the plsr function provided by the pls package in R [29].’

5.       Line 233, ‘The over-representation was based … ’ misses the word ‘analysis’. The sentence should read ‘The over-representation analysis was based … ’.

6.       Line 336, is not fully clear. Were the lower levels of taurine etc. observed in your current study? Or were they in the literature?

7.       Line  344, has a typo. ‘reports’ should read: ‘report’.

8.       Your Statistical Analysis is well done. Would it be an improvement if you introduce a test of independence of the biomarkers and measurements (basal and after 1 week)?  With arbitrary data (made up for the purpose of illustration): such a test could output independence beyond your test of the normality of the data.

9.       Lines 378 – 379, the sentence seems wrong ‘This transport system seems to be under mTOR (mammalian target of rapamicine) …’. Is the word ‘under’ correct here? Why is in ref. [55] MTOR written in full capitals? Is ‘mTOR’ erroneously spelled?

10.   Lines 381, 385, the shortcut ‘(called System A)’ is redundant. You only refer once to it: in line 385.

11.   Line 401, you conclude that a novel placental analysis technique is designed in this paper. But you did not announce this before in the text. The novelty should be explained in the methods chapter and announced in the introduction, I think.

12.   References [7], [10], [29] and [30] are incomplete or wrong. For instance nr. [29] needs pages and has a wrong issue number.  Your reference [30] also misses information: Kuhn, M. (2008). Building Predictive Models in R Using the caret Package. Journal of Statistical Software28(5), 1–26. https://doi.org/10.18637/jss.v028.i05

13.   Your references list is inconsistent. It has many items with and/or without the doi number.

Author Response

Dear reviewer,

thank you for your work on our manuscript and for the interesting and useful suggestions for improvement.

Please find below a list of responses to all the reported concerns.

  1. Lines 33, 35, you say ‘higher’ and ‘lower’. Missing is here: higher to what? and ‘lower than what?

The reason for this remark is that the reader does not necessarily have your prior knowledge on the matter, so she/he does not know your internal mental comparison.

Reply: we reported higher/lower in both SGA and LBW. Due to word rationing in the abstract we cannot add other words. Anyway, the statements let no doubt about the metabolites concentration trend.

  1. Line 132, has a blank between the less than sign and the 10% percentage. This is inconsistent with the other use of this symbol.

Reply: Thanks, eliminated

  1. Lines 205 – 229 do miss illustrations from the research application. To which variables did you apply the principal component analysis? Currently, you only have a general explanation here which does not help to understand to what you apply it. You do not mention which classes were due to overfitting, for instance. For instance: you could refer here, or put here part of your PLS-DA explanation from lines 298 and further.

Reply: I’m sorry I’m a bit confuse because we did not apply PCA and we did not mentioned it in the MS

  1. Line 213, is unclear. ‘… using plsr function provided by R pls package [29].’ Should read ‘using the plsr function provided by the pls package in R [29].’

Reply: Thanks rephrased

  1. Line 233, ‘The over-representation was based … ’ misses the word ‘analysis’. The sentence should read ‘The over-representation analysis was based … ’.

Reply: Thanks added

  1. Line 336, is not fully clear. Were the lower levels of taurine etc. observed in your current study? Or were they in the literature?

Reply: Yes they were reported in our results. We clarify it.

  1. Line 344, has a typo. ‘reports’ should read: ‘report’.

Reply: Reply: Thanks, corrected.

  1. Your Statistical Analysis is well done. Would it be an improvement if you introduce a test of independence of the biomarkers and measurements (basal and after 1 week)? With arbitrary data (made up for the purpose of illustration): such a test could output independence beyond your test of the normality of the data.

Reply: Thanks for the suggestions. Unfortunately, these analyses as now reported were conducted some years ago and so it is not possible for us to make this kind of evaluation. We will use this suggestion in our next project.

  1. Lines 378 – 379, the sentence seems wrong ‘This transport system seems to be under mTOR (mammalian target of rapamicine) …’. Is the word ‘under’ correct here? Why is in ref. [55] MTOR written in full capitals? Is ‘mTOR’ erroneously spelled?

Reply: Thanks, corrected. We changed “under” in “regulated by”

  1. Lines 381, 385, the shortcut ‘(called System A)’ is redundant. You only refer once to it: in line 385.

Reply: we have only extruded the association between the two terms once at the beginning of the paragraph. We have then only used the term system a

  1. Line 401, you conclude that a novel placental analysis technique is designed in this paper. But you did not announce this before in the text. The novelty should be explained in the methods chapter and announced in the introduction, I think.

Reply: The aim of the paper is not to introduce the new sample pretreatment as we already published a paper reporting it. However, as you suggested we introduce the suitability of oven-dried placentas for this kind of study in material and methods section. Due the other reviewers’ suggestion we moved the most part of these sentences in discussion section.

  1. References [7], [10], [29] and [30] are incomplete or wrong. For instance nr. [29] needs pages and has a wrong issue number. Your reference [30] also misses information: Kuhn, M. (2008). Building Predictive Models in R Using the caret Package. Journal of Statistical Software, 28(5), 1–26. https://doi.org/10.18637/jss.v028.i05

Reply: Thanks for your suggestions. We used the Zotero management system for references. We imported the citation using the official tool. However, we will check the references style and consistence in the proofing phase. Thanks for your suggestions.

  1. Your references list is inconsistent. It has many items with and/or without the doi number.

Reply: Please see the reply to the point 12.